# Low Temperature Growth of the Nanotextured Island and Solid 3C-SiC Layers on Si from Hydric Si, Ge and C Compounds

**Lev K. Orlov** [1,2,*]**, Vladimir I. Vdovin** [3]**, Natalia L. Ivina** [4]**, Eduard A. Steinman** [5]**,
Yurii N. Drozdov** [1] **and Michail L. Orlov** [6]

[1]  Institute for Physics of Microstructures, Russian Academy of Science, Nizhni Novgorod 603950, Russia;
   drozdov@ipm.sci-nnov.ru
[2]  Institute of Physical—Chemical Technology & Material Sciences, Alexeev Nizni Novgorod State Technical
   University, Nizhni Novgorod 603950, Russia
[3]  Rzhanov Institute of Semiconductor Physics, Siberian Branch of Russian Academy of Sciences,
   Novosibirsk 630090, Russia; vdovin@gmail.ru
[4]  Mera NN, Nizhni Novgorod 603950, Russia; nataivina@mail.ru
[5]  Institute of Solid State Physics, Russian Academy of Science, Chernogolovka, Moscow distr. 142432, Russia;
   steinman@mail.ru
[6]  Chair of Informatic & Information Technology, Russian Presidential Academy of National Economy and
   Public Administration, Nizhni Novgorod 603950, Russia; orlovm@ipmras.ru
*  Correspondence: orlov@ipm.sci-nnov.ru; Tel.: +7-806-3688823

**Abstract:** Different growth stages and surface morphology of the epitaxial 3C-SiC/Si(100) structures
were studied. Heterocompositions were grown in vacuum from hydric compounds at a lower
temperature. The composition, surface morphology and crystal structure of the 3C-SiC films were
tested using X-ray diffraction, second ion mass spectrometry, scanning ion and electron microscopy,
photo- and cathode luminescence. It was demonstrated that the fine crystal structure of the 3C-SiC
islands was formed by the close-packed nanometer-size grains and precipitated on the underlying
solid carbonized Si layer. Luminescence spectral lines of the solid carbonized Si layer, separated
island and solid textured 3C-SiC layer were shifted toward the high ultraviolet range. The spectra
measured by different methods were compared and the nature of the revealed lines was considered.
This article discusses a quantum confinement effect observation in the 3C-SiC nanostructures and
a perspective for the use of nanotextured island 3C-SiC layers as a two-dimensional surface quantum
superlattice for high-frequency applications. The conductivity anisotropy and current-voltage
characteristics of the two-dimensional superlattices with a non-additive electron dispersion law in
the presence of a strong electric field were studied theoretically. Main efforts were focused on a search
of the mechanisms allowing realization of the high-frequency negative dynamical conductivity for
the structures having a positive static differential conductivity.

**Keywords:** gas phase epitaxy; hydric compounds; 3C-SiC layers; growth stages; luminescence;
quantization; superlattice

---

## 1. Introduction

There is now a heightened interest to the cubic silicon carbide layers grown on silicon substrates
due to the prospects of their use as a compositional element of high-speed silicon electronic devices,
particularly in the design and fabrication of highly efficient wide band gap emitters in silicon diodes [1,2]
and heterotransistors [3,4], and for the application as a high-Q resonator and high-temperature pressure

sensors [5]. Along with a silicon dioxide, silicon carbide layers are actively used in microelectronics as a highly effective chemical and radiation-resistant covering with a higher $SiO_2$ thermal conductivity. These characteristics make SiC films a very promising heat radiator for new design chips with a high active element packing density. Silicon carbide has a large energy gap and elevated light-emitting transmissibility at room temperature in comparison to silicon. Therefore, at present, silicon carbide is considered among the $A^{IV}B^{IV}$ materials as the only alternative to large energy gap semiconductor binary compounds based on metal oxides and nitrides when ultraviolet (UV) sources must be created.

Big difficulties under growing the high-quality planar single heteroepitaxial 3C-SiC/Si structures with a large lattice mismatch (~20%) are stimulus to study the growth mechanisms, surface morphology, and interface characteristics of the epitaxial carbide layers. Morphological features of the 3C-SiC surface are the result of the excess carbon on a surface and a strong in-plane heterogeneity associated with the elastic deformation fields [2]. Island morphological structures are often regarded now as quite an independent type of microelectronic elements with their own unique electronic properties. Separate interest is associated with the general problem of creating ordered two-dimensional arrays of quantum nanodimensional objects (quantum dots) in bulk and on the surface of crystals, which are promising for various applications of micro- and nanoelectronics [6]. In this aspect, heteroepitaxial structures based on crystalline materials with strongly different lattice constants, which are in particular silicon and its diverse carbide phases, seem to be appealing because it is possible to form on their base various, often very specific, layered and island mono- and polycrystalline compositions involving nanoscale elements exhibiting non-trivial properties [7,8].

On the other hand, a significant focus for a long time has been given to finding the best growth conditions of the two-dimensional nanoisland arrays. Among these systems, the Ge/Si heterostructures grown by the Stranski–Krastanow mechanism received the greatest development [9]. Not less interesting structures are 3C-SiC/Si heterostructures with the significantly different lattice parameters of the contacting materials. Multiform layered and island structures exhibiting non-trivial properties may be formed on the base of the silicon carbide layers [10–13].

The goal of this work was to produce the respective structures using simultaneously both the low-temperature 3C-SiC film growth mode [12] and a large lattice mismatch of the Si/3C-SiC heteropair. Island 3C-SiC layers have been grown on the preliminary carbonized Si(100) surface using silicon carbonization specific at decreased growth temperatures and hydrogen-containing compounds as silicon and carbon sources. One of the interests was also to study the effect of surface stresses on the shape of island formations at the initial stage of growth and to find the conditions for the appearance of nanoislands in the form of either individual single crystallites or textured polycrystals on the growth surface. In this paper, the surface morphology of the 3C-SiC/Si(100) structures has been studied at different stages of their formation with a use of the light interference microscopy, ion/electron scanning microscopy and different luminescence methods. Other objectives of this work were to establish the real reasons for the light-emitting ability of silicon carbide layers within a broad range of wavelengths and the study of the possibilities for observation of effects associated with the spatial confinement of charge carriers within silicon carbide microcrystallites. It was necessary to understand both photoluminescence mechanisms in the structures and other characteristics of the low-temperature-grown carbide layers. The most interesting from them were the problems of the carbon solubility in germanium, of the spatial electron confinement [14] in SiC nanocrystals and the observation possibility of the quantum superlattice [15] properties in the SiC surface nanostructures.

## 2. Experimental Details: Technology, Crystal Structure, Composition and Morphology of the Silicon Carbide Layers

Silicon carbide layers were grown in a vacuum chamber (base pressure ~$3 \times 10^{-7}$ Torr) on Si substrates with a different orientation. Substrates were cleaned in a hydrofluoric acid vapor and annealed about 30 min at 1200 °C temperature in the growth chamber under low (~0.01 mTorr) silane pressure. The preliminary carbonization of silicon surface at temperature 1000 °C was carried out from

hydrocarbons by the chemical conversion method performed traditionally via the substitution of Si atoms in the silicon crystal lattice by carbon atoms [12,16]. This method ensures a growth of the 3C-SiC layers with an acceptable thickness (few tenths of a micrometer) even at one source of hydrocarbons. The further growth of the cubic phase of silicon carbide layers on silicon was performed at reduced 600–900 °C temperatures with the use of a silane and hexane mixture under the partial gases pressure from 0.01 mTorr to 3 mTorr [17]. In a small number of experiments, we used also germane ($P_{GeH4}$~$P_{SiH4}$) as an additional gas source to study the possibilities of Ge atom embedding in 3C-SiC crystal lattice and the features of a Ge accumulation on a layer surface. The low heteroepitaxy temperature was used to determine the minimum possible temperature of carbide formation in silicon films, on the one hand, and to obtain the minimum (nanometer scale) grain size in the grown islands forming a continuous carbide film, on the other hand. Increase in the substrate temperature and the pressure of gases in the reactor promoted the nucleation of a greater number of islands on the growth surface, and their expansion further resulted in the formation of a continuous microcrystalline film. The formation of a silicon carbide layer, even under the conditions of decreased growth temperatures ($T_{gr} < 800$ °C) at a relatively high hydrocarbon vapor pressure in the reactor, was confirmed by XRD (Bruker), electron diffraction (EED), electron (Carl Zeiss SMT;) and He ion microscopy (IRS-NT), second ion mass spectrometry (SIMS), photoluminescence (PL), and cathodeluminescence (CL) methods [17–19].

In this section, we study the surface morphology of the grown carbide film and focus attention on the stages of nucleation and growth of separate islands as well as formation of the continuous layer. Understanding the growth mechanisms and peculiarities of the silicon carbide layer structure is very important due to the attractiveness of the 3C-SiC epitaxial films for practice. For the growth method used, the growing of the SiC layer is supposed to differ [19] from the layer growing in traditional Ge/Si system characterized by the Stranski–Krastanow mechanism. A comparative analysis of the 3C-SiC layer surface morphology is presented for the samples grown at different growth temperatures (650 °C < $T_{gr}$ < 950 °C) but identical other conditions of the technological experiment ($t_{gr}$ = 180 min, $P_{C6H14}$ = 1 mTorr; $P_{SiH4}$~$P_{GeH4}$ ~0.3 mTorr, $d_{SiC}^{max}$ ($T_{gr}$ ~950 °C) ~300 nm). The growth temperature determines the disintegration rate of the hexane ($CH_3$) [11,16] and silane ($SiH_3$) [20] molecule fragments adsorbed by growth surface and consequently a number of silicon and carbone adatoms participating in a growth process. The study of the surface morphology of structures obtained at the same molecular flow rates but at different substrate temperatures shows that the surface morphology of the carbide film changes continuously during the growth process. The 3C-SiC film growth begins with the formation of a thin continuous silicon carbide layer due to the carbon diffusion in the direction of a normal to a layer plane [21]. The follow-up layer growth was performed at temperatures from 650 °C to 950 °C with a silane, germane and gexane mixture. Individual microcrystalline formations increased the size from ten nanometers at the base of growth figures to several microns during the epitaxial growth of the film. At the same time, the nucleation of new islands with the resulting increase in their density can be observed on the growth surface. The island nucleation and expansion rates are governing by the velocity of the arrival of silicon and carbon atoms to the epitaxial surface. On Figure 1, one can observe the different stages of SiC epitaxial growth on the thin monocrystaline 3C-SiC underlayer from the island nucleation to the continuous layer formation. At the island nucleation stage, the separated islands have the orientation along the <110> crystallography direction and nanocrystal structure. The photos obtained for individual islands by helium (He) ion microscopy (Figure 1d), scanning probe and electron microscopy demonstrate also the existence of a finer crystal structure for each microcrystalline island. Every individual 3C-SiC island is a textured polycrystal with a nanometer grain size (of less than 30 nm) in the film plane. At present, however, the characteristic size, at which the transition from one mechanism of growth with the formation of the single-crystal structure of an island to the other columnar mechanism with the formation of a textured polycrystal in the volume of an island may occur, remains unknown for an island nucleating on the growth surface. As the layer thickness increases, its structure transforms into the micron-sized large-block structure with the cavities at the boundaries between joined islands (Figure 1c). Further growing leads to the formation of

the continuous layer consisting of the joined islands that conserve their initial morphology (insert on Figure 1b). As a result, relatively thick (>0.1 µm) silicon carbide films have a texture polycrystalline structure with the grain size of nanometer scale (inserts on the Figure 1c). At the same time the surface morphology of these thick films has an acceptable perfection with the surface roughness $S_q$ below 10 nm.

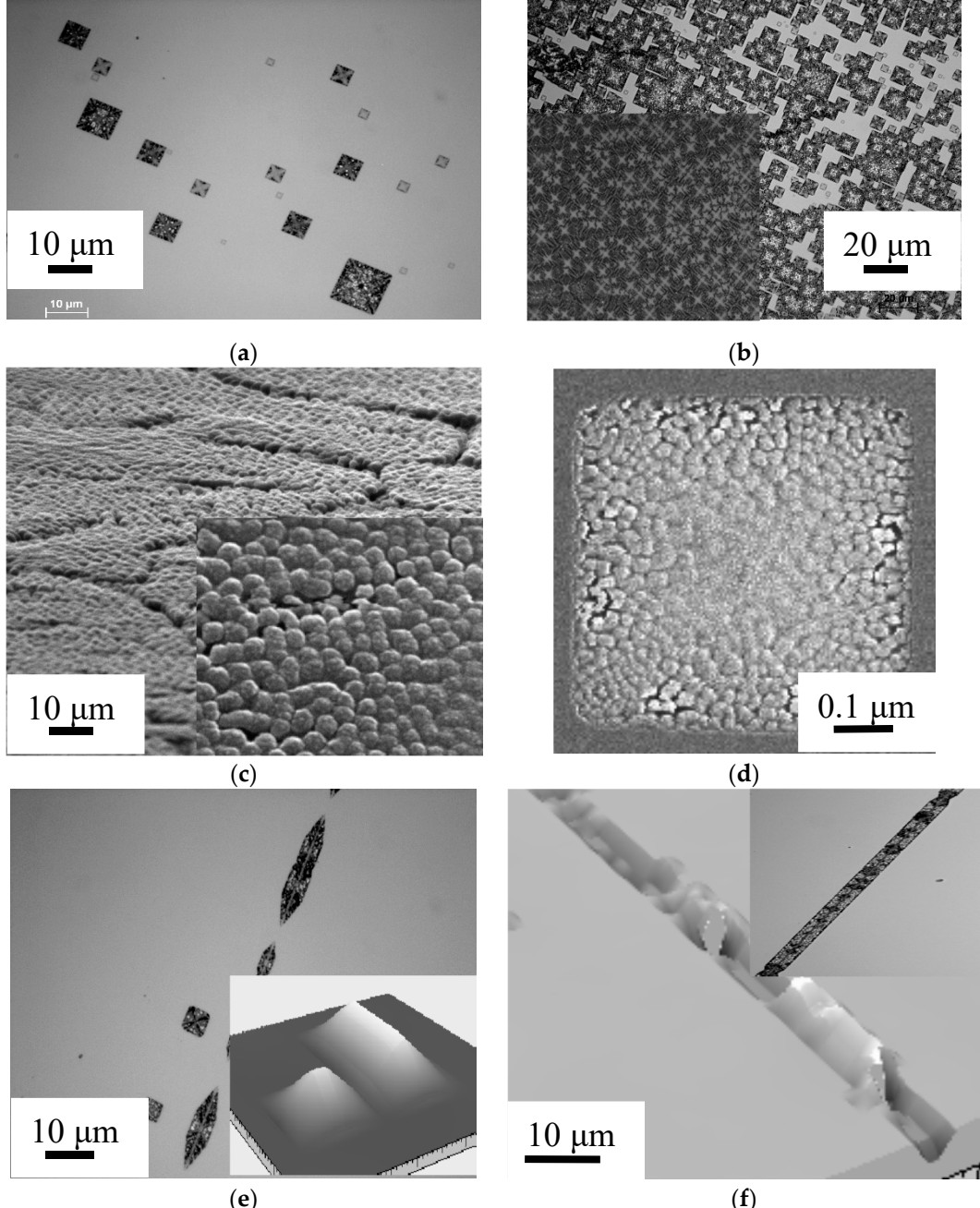

**Figure 1.** Micrographs (a,b,e,f—light microscopy; c,d—ion (He) microscopy) of 3C-SiC(100) layer surface at the different growth stages: (**a**) nucleation of the separate islands on the carbonized Si surface; (**b**) multiple coalescence of islands with conservation of island morphology (insert—thick continuous layer); (**c**) thick continuous layer with the large-block structure (insert—grain structure of the separate block); (**d**) separate island with the grain structure; (**e,f**) optical microscopy images of the dotted (**e**) and linear ((**f**)—along the line of the crystalline plane slippage) morphological formations on the surface of 3C-SiC/Si(100) heterostructures.

The further growth of a microcrystalline island due to the accumulation of the elastic stresses in it can lead, however, to the disorientation of individual atomic planes and the formation of grain boundaries instead of the appearance of the misfit dislocations as it takes place, e.g., in a Ge/Si system [22]. In the stressed 3C-SiC/Si(100) system, more complicated growth figures may be observed on the layer surface. They appear generally along the slip lines of crystalline planes, which arise during the long annealing (Figure 1e,f are obtained for sample grown under the next conditions: $T_{gr}$ ~650 °C, $t_{ann}$ = 70 min, $t_{gr}$ = 280 min, $P_{C6H14}$ ~0.1 mTor, $P_{SiH4}$~$P_{SiH4}$ ~0.09 mTorr) and associate with the outcrop of mismatch dislocations. Wave light interference (WLI) analysis of the point formations on the surface of the 3C-SiC layer shows a rather complex three-dimensional shape of the islands in the strained structure (Figure 1e). The linear growth figures (Figure 1f) have a V-shape form across their guides. Inhomogeneous distribution of the adsorbed molecules along a V-shape groove provides a complex profile of a linear growth figure along its guide.

The component distributions across the 3C-SiC/Si structures were studied by secondary ion mass spectrometry (SIMS). The typical SIMS profiles are shown in Figure 2b. The thickness of the homogeneous silicon carbide layer was determined from the carbon ($^{12}$C) or silicon carbide (SiC) concentration declination and from the silicon ($^{28}$Si) concentration growth in the SIMS spectra. Carbon atoms diffuse into the silicon substrate for the length more than 3C-SiC layer thickness, forming an intermediate $Si_{1-y}C_y$ solid solution layer between the 3C-SiC film and Si substrate. Figure 2b shows also the distributions of oxygen ($^{16}$O) and germanium ($^{74}$Ge) in one of the 3C-SiC/Si structures grown with the use of the germane source. The maximum concentrations of oxygen and germanium atoms were observed in the structure under the carbide layer, which is likely an effect of poorer solubility of oxygen and germanium in the silicon carbide. During SiC layer growing, oxygen and germanium atoms are displacing into the Si substrate forming under the 3C-SiC film an $Si_{1-x}C_x$:Ge sublayer enriched by oxygen. The appearance of oxygen atoms with the formation of $SiO_2$ inclusions in the growing carbide layer alongside with nitrogen atoms can be explained by the presence of a high level of residual atmospheric gases in a reactor of a used setup. The observed accumulation of oxygen near the heterointerface (curve 5 on Figure 2b) is due to the effect of oxygen removal from the 3C-SiC layer at the stage of the surface carbonization of a silicon plate. The concentration of hydrogen in the layers remains lower than the oxygen concentration, so no complete passivation of broken oxygen bonds takes place in the structure.

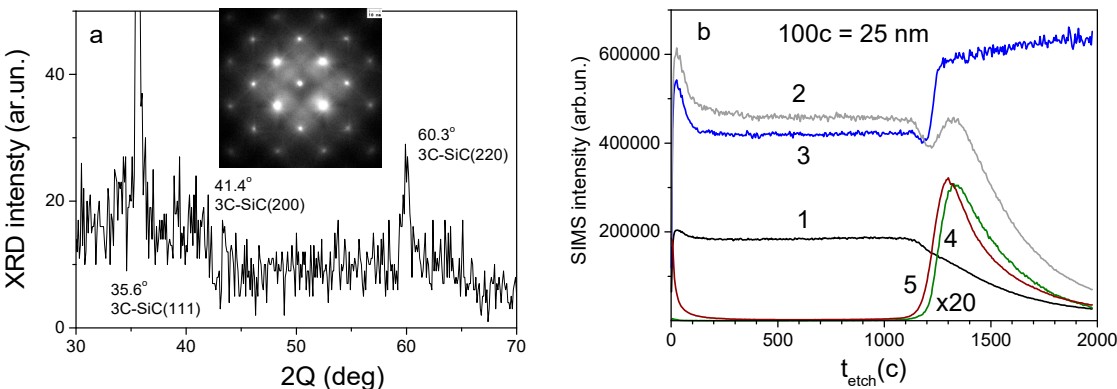

**Figure 2.** X-ray diffraction and electron diffraction patterns (**a**) and distribution according to secondary ion mass spectrometry (SIMS) data (**b**) of main elements in the growing carbide layer (SiC (1), $^{12}$C (2), $^{28}$Si$_2$ (3), $^{74}$Ge (4), $^{16}$O (5)) for the sample corresponding Figure 1b.

The character of the germanium component distribution is of interest in the considered structure. Despite germanium being fed to the growth chamber during the whole growth of the carbide film, its concentration in the 3C-SiC layer was insignificant, which corresponds to the generally accepted condition of poor solubility of germanium in silicon carbide. Although the segregative accumulation

of germanium occurred on the surface, it was very insignificant in comparison with the total amount of germanium atoms diffused into the structure depth to the heterojunction region. The main number of germanium atoms incorporated into the structure lattice was located due to the gettering effect. As seen from Figure 2b (curve 4), in vicinity of the heteroboundary from the side of the Si substrate there is a formed solid solution sublayer which provides, in a number of cases, the increased abruptness of the formed heterojunction [23]. We observed, for the first time, the "membrane" effect that is a penetration of germanium atoms adsorbed on the growth surface during the germane pyrolysis process through the growing layer of silicon carbide to its inner heteroboundary and the formation of a thin layer of SiGeC triple solid solution of a variable composition in vicinity of the SiC/Si heterojunction. A probable mechanism of the membrane effect observed is the accelerated diffusion of germanium atoms over grain boundaries, which is directed to the internal boundary of the SiC layer. No appreciable accumulation of germanium atoms inside the polycrystalline silicon carbide film was observed here.

## 3. Light-Emitting Properties

Among $A^{IV}B^{IV}$ materials, at present silicon carbide is considered the only alternative to large energy gap semiconductor binary compounds based on metal oxides and nitrides when UV sources must be created. As a rule, the heavy elastic strains observed in the layers promote the formation of films with a nanocrystalline structure. This phenomenon can in turn lead to the appearance of new properties of the material. These properties are often associated with the effect of the spatial limitation of electrons within nanocrystallites of the system. Different approaches have therefore been proposed in the literature to modify the crystalline structure of the layers, and composite materials with nanocrystallites have been produced using various techniques and a choice of different processing conditions [24–29]. A decrease in the crystallite size by using electrochemical methods or choosing the corresponding technological growth conditions allows one to observe the photoluminescence with an increased emission efficiency in 3C-SiC films in the UV spectral region [8], including the photoluminescence associated with the manifestation of the quantum confinement effect in the electronic spectrum of nanocrystalline structures [14]. These findings have rekindled the interest of many researchers in the light-emitting properties of different silicon carbide modifications.

The light-emitting properties of the polycrystalline-textured cubic silicon carbide films on silicon substrates were investigated by us in the range from room temperature to liquid helium temperature. The electronic subsystem of the 3C-SiC layer was usually excited by a helium–cadmium laser operating at a wavelength of 325 nm (3.81 eV) and a power of 5 mW. The use of a BS7 glass filter made it possible to cut off the excitation line and to analyze the observed features in the photoluminescence spectrum in the wavelength range above 350 nm. The observed relatively broad luminescence band (curves 2–4, Figure 3a,b) is characterized by the maximum shifted with respect to the absorption band edge of the cubic silicon carbide phase toward the ultraviolet region of the optical frequency range. The photoluminescence in silicon carbide layers in the wavelength range 400–500 nm was studied by many investigators [7,8,27–31]. The measurements performed in the majority of works were focused mainly on the observation of the radiated recombination in films at room temperature (curves 1(a), 4_1(b) Figure 3). The use of the wide range of measurement temperatures in our experiment enabled us to reveal a fine structure of the band in the photoluminescence spectrum [18] and to elucidate the influence of the substrate orientation or sample number (curves 1–3 Figure 3a), and temperature factor (curves 4_1–4_3 Figure 3b) on the position and width of the luminescence bands. In addition to the measurements of the structural and phase characteristics of the films under investigation, this allowed us to draw certain conclusions about the possible nature and mechanisms of light-emitting recombination in polycrystalline layers of cubic silicon carbide. The analysis of the spectral curves (Figure 3a,b) has demonstrated that the relatively broad luminescence band observed in the experiment in the energy range from 2.4 to 3.5 eV is split into a set of Lorentzian components. The position of the spectral lines observed in the long wavelength ($h\omega_{PL} \approx 2.4$ eV) range corresponds

to the luminescence spectra typical for cubic silicon carbide [25,27–29]. The presence of this layer is confirmed by the cathode luminescence spectra (Figure 3c) both from separate islands and smooth regions in images (Figure 1a) [21], too.

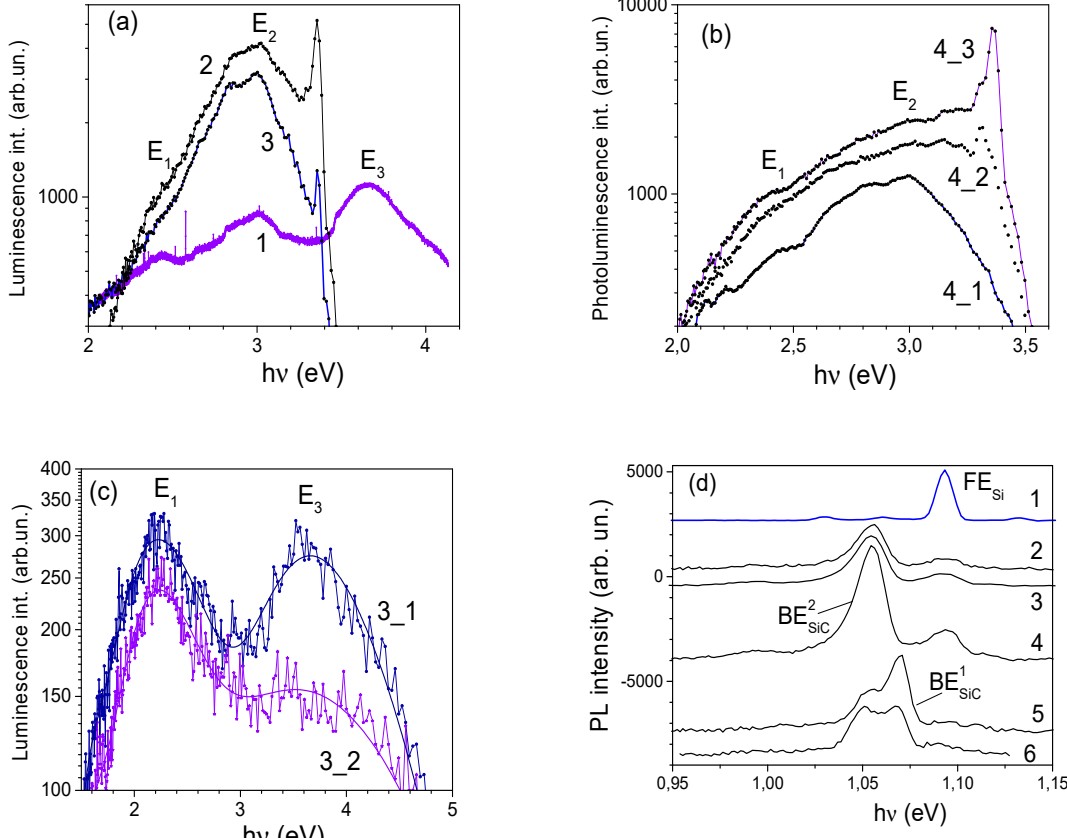

**Figure 3.** Photoluminescence spectra (**a**,**b**) from (sample 1,2) 3C-SiC/Si(111) (curve 1,2) and (sample 3,4) 3C-SiC/Si(100) samples obtained at room (curves 1,2,4_1), 80 K (curve 4_2) and 10 K (curves 2,3,4_3) temperatures with the use of the third harmonic titan-sapphire laser (curve 1) and helium-cadmium (2,3,4_i) laser. Cathodoluminescence spectra (**c**) of the sample 3: curve 3_1 from surface of the individual 3C-SiC(100) island and curve 3_2 from a surface of the solid carbonized 3C-SiC sublayer**.** Near infrared spectra (**d**) at a temperature of 6 K for the Si layer (1) and for the 3C-SiC/Si(100) structures grown from hexane (5), from hexane and silane (2,4) and from hexane, silane and germane (3,6) under similar other conditions.

The most interest in the literature has been expressed in the study of the ultraviolet photoluminescence of silicon carbide films in the energy range $h\nu \approx 2.8$–3.0 eV. The photoluminescence observed in silicon carbide layers containing nanostructured elements in this spectral range also has a significant intensity at room temperature, which is of great practical interest. The emission in the optical region in the vicinity of 3 eV was observed for porous silicon carbide films [24,25] and silicon carbide layers with a nanocrystalline structure [27–29]. Similar spectra were recorded by us at room temperature upon excitation of the electronic subsystem of the structure with the other methods: with a third harmonic $\lambda_{excit}$ = 266 nm (4.66 eV) of the femtosecond titanium–sapphire laser (curve 1, Figure 3a) and with a Helium (He) beam from a scanning ion microscope (Figure 3c). The 3C-SiC layer (1) with the thickness ~2.5 mm was grown on Si(111) substrate from hexane ($P_{C6H14}$ ~0.03 mTorr) at temperature $T_{gr}$ ~900 °C. The spectra 2(3) were measured for the 3C-SiC/Si(100) structures grown at temperature $T_{gr}$ ~900 (600) °C with the gas pressure $P_{C6H14}$ ~0.03 (1) mTorr, $P_{SiH4}$ = 0.005(0.3) mTorr, $P_{GeH4}$ = 0.003(0.3) mTorr in a growth chamber, accordingly.

The photoluminescence spectrum at room temperature (curve 1, Figure 3a and curve 4_1, Figure 3b) contains three pronounced rather broad bands with energies of 2.44, 3.00, and 3.66 eV at the maxima [24]. The cathodeluminescence (CL) spectra recorded at room temperature from different areas of the 3C-SiC/Si(100) structure surface in the neighborhood of an individual island contain the major band with a maximum at an energy $E_g$(3C-SiC) $\approx$ 2.25 eV. This line is characterized by strong broadening, which is most likely due to nonuniform strains in the plane of the structure, and present in the CL spectra from both the single-crystal carbidized silicon substrate layer and different areas on the surface of an individual microisland [21]. The shift of this photoluminescence band towards the near-ultraviolet region was absent due to the high temperature of performed measurements.

In the vast majority of the aforementioned works [7,8,24,29–33], the photoluminescence line in the range with the $E_2$ energy was attributed to a shift toward the ultraviolet spectral region due to the manifestation of the quantum confinement effect in the energy spectrum of nanocrystals involved in the composition of the 3C-SiC/Si heterostructure. A decrease in the size of silicon carbide crystalline particles involved in the composition of the film has to lead to a shift in the maximum of the photoluminescence spectrum toward the short-wavelength range due to the spatial confinement. However, until recently, reliable experimental data that would confirm the manifestation of quantum confinement effects in nanocrystallites of the system at room temperature had not been available in the literature. None of the publications that we know of (see, for example, the review [33]) reported on explicit dependences of the photoluminescence line shift on the size of the nanocrystallites involved in the composition of the sample. Moreover, other mechanisms may be proposed to explain the observed higher PL energy bands [18,34–36]. It may be, for example, the inclusions of the other carbide phases in the layer structure, but a low temperature growth (<1000 °C) decreases the possibility of the appearance of the high temperature (6H-and 4H-SiC) phases [30]. However, there is some probability that $E_2$ peak is connected with rhombohedric 21R(1010)-SiC microcrystalline phases. This fact follows in particular from XRD measurements: XRD peak with 2θ $\approx$ 37.4° (Figure 2a). Appearance of the rhombohedral phase in the thick 3C-SiC layers may be connected most probably with both the long duration growth process and with the presence in the system of a higher elastic strain. This phase is absent at the first growth stages of a thin solid or island layer that we can see from cathodeluminescence spectra (Figure 3c) when a peak $E_2$ is very small.

The spectral peak is positioned in the far UV radiation region ($E_3$ ~3.66 eV) and can be detected with the excitation sources [21,24,32] different from Ar or He–Cd laser. The emission band $E_3$ was observed for the first time in the luminescence spectra of $SiO_2$ layers containing (via the implantation of Si and C ions) silicon carbide nanocrystallites of 1–2 nm in size [37]. The different possible mechanism of excitation of the luminescence in the UV range of the optical frequency region is considered in the works [21,32,33,37]. The comparative analysis of the behavior of the lines in the observed luminescent spectra with the SIMS data (Figure 2b) demonstrated their obvious dependence on the content of oxygen at the interface between the 3C-SiC layer and the silicon substrate. That allowed us to propose that appearance of the line at 3.66 eV can be assigned to the emission of oxygen vacancies in silicon dioxide microinclusions in the 3C-SiC/Si heteroboundary.

Photoluminescence method is the direct technique to reveal not only the defect structure of the SiC layer, but a damaged point and line defect nearest the interface and layer composition on the silicon side in the vicinity of the heterojunction. In [21], we showed that appearance of the lines at 3.66 eV and 1.55 eV in CL spectra can be connected with the emission transitions in 3C-SiC/Si heterostructure with a participation of the oxygen centers. Most frequently, the luminescence method has been used in the near infrared region to reveal a plastic deformation in elastic strain heterostructures based on silicon and germanium [38], in particular. The typical spectral dependences observed in the near infrared region for a number of 3C-SiC/$Si_{1-y}C_y$/Si(100) structures grown from hexane (5), hexane and silane (2,4) and hexane, silane and germane (3,6) under other conditions are shown in Figure 3b. These dependences were obtained at a temperature of 6 K with the use of a gallium arsenide laser for excitation of the electron-hole subsystem [19]. From these spectra, we can see above all that a relaxation

of elastic stresses accumulated in the system almost entirely occurs along the grain boundaries within a nanocrystalline silicon carbide layer. The influence of the formed intermediate $Si_{1-y}C_y$ alloy layer manifests itself as a well-pronounced change in the band-edge silicon photoluminescence spectrum (Figure 3b). For all the structures studied, we observed, in addition to standard free exciton ($FE_{Si}$) line, the additional bond exciton $BE^{1,2}_{SiC}$ spectral lines having a different temperature behavior of their intensities. One $BE^2_{SiC}$ (~1.055 eV) spectral line can be associated with the electron transitions in the $Si_{1-y}C_y$ (y ~1–2 at.%) layer with the participation of the optical phonons. Another $BE^2_{SiC}$ (~1.07 eV) peak can be associated with the band–band phonon transitions through an intermediate state of local defects in the $Si_{1-y}C_y$ layer. In our case, these defects are most likely defects of the vacancy type. Their formation, as is well known, can be associated with a re-evaporation of silicon atoms during the growth of the intermediated $Si_{1-y}C_y$ alloy layer and a nucleation of voids in the vicinity of the 3C-SiC/Si heterojunction [7,14].

## 4. Discussion

In the first part, we for the first time showed the existence of the fine nanotextured structure of the 3C-SiC island formed on the solid surface carbonized Si layer and generated a natural two-dimensional superlattice. The second part of the paper is devoted to the problem of the observation in this superstructure; some phenomena which may be connected with the manifestation of the quantum confinement effect. The observation of the electron spectra quantization and tunneling effects in the ordered nanocrystal (quantum dot) array is the first step in a way to solve the problems of the two-dimensional quantum superlattice realization. Obviously, a use of the optical methods only for the investigation of the electron states and for a revelation of the unique futures of electron spectra in periodic heterocompositions is insufficient. To take special attention toward the study of the transport phenomena in the considered structures, we discuss in the third part some nonlinearity peculiarities of the electrical characteristics for the two-dimensional quantum superlattices.

Size quantization and tunneling effects in one-layer (quantum well) and multi-layer (superlattices (SL)) semiconductor structures have been actively studied in literature for a long time [39–41]. It is stimulated by the progress in the epitaxial growth technology of semiconductor hetero- and nanostructures on the base of both traditional and various types of strained systems. The tendency toward an island mechanism of layer growth not only promotes the formation of crystalline structures with the ordered arrays of the quantum dots, but experience shows that use of this mechanism might one day help us to create silicon surface, 2D, periodic, close-packed structures with a nanometer size of grains that form a natural two-dimensional surface superlattice (2DSL). This is especially topical for a number of carbide phases due to the quantum superlattice effects that are already observed experimentally in the single carbide layers [15]. Proving quantum phenomena in them is the result of the existence of additional periodic potential in the structure due to the additional long-period rotational symmetry axes. Earlier, multilayer structures with a one-dimensional SL were highly studied [39,41]. In recent years, the attention has been taken also to the two-dimensional and three-dimensional SLs. We showed (see Figure 1d) that two-dimensional surface superlattices may be, for instance, formed in the plane of the SiC nanocrystalline layer of cubic symmetry. Therefore, one may expect that the presence of the additional nanostructure in the microcrystals of the carbide layer will permit to observe the new phenomena not only on the electron diffraction pattern [17,42] but also in optical and electrical characteristics. Below, we should like to demonstrate the differences of the transport properties of the one- and two-dimensional superlattices and consider the more important effects, which could be used for a diagnostic of the quantum states in the structures and for the further practical applications. A proof of a quantum nature of the observed effects and also a prediction of the new phenomena in the considered carbide structures gives a hope to use these heterostructures both for the improvement of the traditional diode and transistor characteristics and also for a development of the new quantum devices such as a Bloch generator or submillimeter quantum cascade laser working at room temperature.

In this section, for instance, we studied theoretically the conductivity anisotropy and volt-ampere characteristic peculiarities that maybe revealed in principle in the lateral quantum two-dimensional superlattice (2DQSL) in the presence of a strong electric field E. This task is very important due to the problem of the Bloch generator in a semiconductor quantum superlattice (SL) [39,43–45]. The observation of the high-frequency (HF) negative differential conductivity (NDC) in quantum SLs remained, until recently, very problematic due to the development of the low-frequency instabilities. Therefore, the efforts of many investigators are focused on the search of the mechanisms allowing the realization of the HF NDC on the Ampere—Voltage (I-V) characteristic sectors with the positive static differential conductivity [40,45].

The calculation method is based on the nice calculation of the Boltzman equation in the strong electric fields that was elaborated by us in the seventies. The complete calculation scheme of the static conductivity for a two-dimensional quantum superlattice with a simplest harmonic dispersion law has been carried out in [46–48]. That allowed us not only to predict the shape of the current voltage characteristics but also to reveal some specific phenomena associated with the system anisotropy. In this paper, we considered the peculiarities of the electron transport in a 2DQSL with the more complete non-additive electron dispersion law:

$$\varepsilon(k_\perp) = \Delta_1\{1 - [\Delta_{11}\cos(k_1 d_1) + \Delta_{12}\cos(k_2 d_2)]/(\Delta_{11} + \Delta_{12})\} + \Delta_2\{1 - \delta_0 \cos(k_1 d_1)\cos(k_2 d_2)\}, \quad (1)$$

where $\varepsilon(k)$ and k are the energy and quasi-wave vector of an electron, $k_i$ (I = 1,2,3) are its components, $\Delta_{ij}$ = const, $d_i$ is 2DSL period along an axis (i). A choice of a more complicated dispersion law for the electrons is connected with the multivalley spectrum of the electrons in the 3C-SiC layers. The calculation of the current voltage (I-V) characteristics in a one miniband approximation was carried out using a kinetic Boltzmann equation for both square ($d_1 = d_2$) and rectangular 2DSL ($d_1 \neq d_2$) (see Figure 1d,e) with a $\tau$-approximation of the collision integral [48]. Exposing this model, we find the current dependences for the main directions in a 2DSL:

$$j_1 = j_0 \, (E/E_0) \, \cos\psi \, \{D_{11}F_{10}/[1 + (E/E_0)^2\cos^2\psi] + F_{11}D_{20}[1 + (E/E_0)^2(\cos^2\psi - \chi\sin^2\psi)] \, / \\ [1 + 2(E/E_0)^2(\cos^2\psi + \chi\sin^2\psi) + (E/E_0)^4 \, (\cos^2\psi - \chi\sin^2\psi)^2]\}, \quad (2)$$

$$j_2 = j_0 \, (E/E_0) \, \sin\psi \, \{D_{12}F_{01}/[1 + \chi(E/E_0)^2\sin^2\psi] + F_{11}D_{20}[1 - (E/E_0)^2 \, (\cos^2\psi - \chi\sin^2\psi)] \, / \\ [1 + 2(E/E_0)^2(\cos^2\psi + \chi\sin^2\psi) + (E/E_0)^4 \, (\cos^2\psi - \chi\sin^2\psi)^2]\}, \quad (3)$$

Here $j_{1(2)}$ are components of a current density in the direction of principal axes of 2DSL, $j_0$ = $eF_0/2\pi hd_2$, $\chi = (d_2/d_1)^2$, $\psi$ is the angle of a field $E = h\Omega/ed_\psi$ deviation from principle axis ($x_1$), $E_0$ = $h/e\tau d_\psi$, $d_\psi = d_1/\cos(\psi)$, $\Omega$—Bloch oscillation frequency. $D_{20} = \delta_0\Delta_2$, $D_{10} = \Delta_1/\{(\Delta_{11} + \Delta_{12})\}$, $D_{11}$ = $D_{10}\Delta_{11}$, $D_{12} = D_{10}\Delta_{12}$, $Y_{ij} = D_{ij}/k_BT$. The matrix elements $F_{\nu\mu}$ for the Bolzman distribution are equal:

$$F_{\nu\mu} = F_0 \mathrm{Re} \int_0^{2\pi}\partial x_1 \int_0^{2\pi}\partial x_2\{\exp(i\nu x_1 + i\mu x_2)\exp[(Y_{11}\cos(x_1) + Y_{12}\cos(x_2) + Y_2\cos(x_1)\cos(x_2)]\}. \quad (4)$$

The augend in curly brackets in the formula (2) corresponds to the well-known expression for I-V characteristics of a 1DSL with a NDC at $\Omega\tau > 1$. The addends in Equations (2) and (3) are determined by the non-additive addend in the dispersion law (1).

Earlier, the I-V characteristics of a 2DSL with an additive dispersion law were studied in [46–48]. The conditions for their appearance on the additional displacement of the NDC toward high or low fields from the critical field of nonlinearity [48] and additional damping of Bloch oscillation [47] have been found. In 2DQSL, the deviation of the electron dispersion law from harmonious has a noticeable effect on the form of the I-V characteristics in a strong electric field. Our calculations (Figure 4) show that the addends in (1) can result in a change of the I-V dependency. The appearance of the additional wide maximum in the high electric fields is connected with the hitting and redistribution of the charge carriers between different miniband valleys and possibly with the formation of some Stark ladders in

the lowest miniband [49]. It is principle if the current voltage characteristic of the SL contains more than one curve lines with a positive and negative differential conductivity [40]. In this situation, there is a chance to realize in a system the negative dynamic conductivity is absent during low-frequency self-excited oscillation [45].

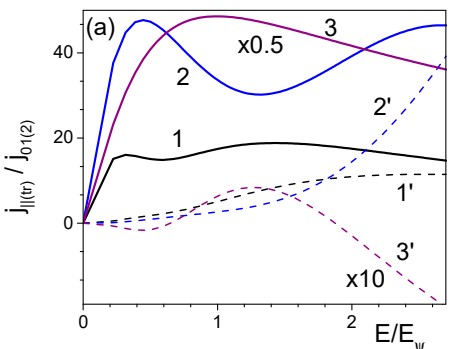 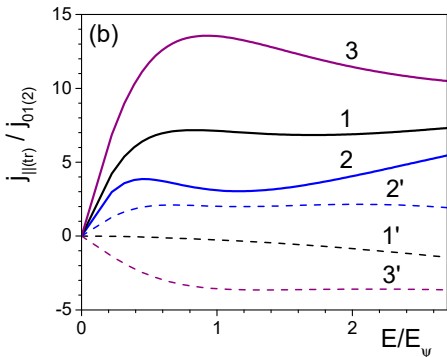

**Figure 4.** Dependences longitudinal $j_\parallel$ (curves 1–3) and transversal $j_{tr}$ (curves 1′–3′) current components on electric field $E/E_\psi$, in a direction angularly (**a**) $\psi = \pi/4.5$ (1,1′), $\pi/6.5$ (2,2′), $\pi/10$ (3,3′) to the field direction for parameter values are equal: $\Delta_1 = \Delta_2 = 5$ meV (curves 1,1′), $\Delta_1 = 2$ meV, $\Delta_2 = 10$ meV (curves 2,2′), $\Delta_1 = 10$ meV, $\Delta_2 = 5$ meV (curves 3,3′), $d_2/d_1 = 2$, $k_BT = 7$ meV, $\Delta_{11} = \Delta_{12} = 1$; (**b**) $\psi = \pi/10$, $d_2/d_1 = 2.7$, $\Delta_1 = \Delta_2 = 5$ meV, $k_BT = 20$ meV, $\Delta_{11} = 1$, $\Delta_{12} = 1$ (curves 1,1′), $\Delta_{11} = 1$, $\Delta_{12} = 10$ (curves 2,2′), $\Delta_{11} = 10$, $\Delta_{12} = 1$ (curves 3,3′), $\delta_0 = 1$.

## 5. Conclusions

Nucleation and expansion of 3C-SiC islands on the carbidized silicon surface and a formation of the continuous 3C-SiC layer during a low temperature epitaxial process with the use of the hydric compounds have been traced. The main attention is focused on the investigation of the composition, microstructure, and surface morphology of the layers. The effect of structural ordering forming the growth figures on the epitaxial surface has been studied for the 3C-SiC/Si system. The grain-oriented nanostructure of developing microislands and the effect of the surface strains on island shape and other morphological defects were demonstrated by us for the first time.

The nature and mechanisms of the light-emitting properties of these structures are studied in detail. The comparative analysis of the behavior of some spectral lines in the observed luminescent spectra demonstrated their obvious dependence on the content of the oxygen in the layers of the 3C-SiC/Si$_{1-y}$C$_y$/Si structure. The structure of the heterojunction formed during the growth of the 3C-SiC layer and formation of the intermediate alloy layer have been studied using the results of the photoluminescence investigations in the near-infrared wavelength range and the data obtained from the mass-spectrometric analysis. The study of the light-emitting properties of the structures in the band-edge photoluminescence showed that concentration of point defects in the Si$_{1-y}$C$_y$ alloy layer significantly increases but the dislocation structure is not well pronounced.

A perspective of the manifestation in these heterostructures the quantum confinement effect and the possibility to consider these systems as the two-dimensional quantum superlattices were discussed. We considered the anisotropy of conductivity and the features of the current voltage (I–V) characteristic for different directions of the current and electric field (relative to the axes of the superlattice) applied to the two-dimensional quantum superlattice (2DSL) with an electron dispersion law more complex than the simple harmonic. It is shown that the deviation of the electron dispersion law in 2DSL from the harmonic in a strong electric field (i.e., miniband structure) has a noticeable effect on the form of the I–V characteristic of 2DSL. The appearance of several peaks on the I–V characteristic in a strong field is due to mixing of the Stark electron levels for orthogonal directions and the Stark electron levels from different valleys.

**Author Contributions:** Methodology, L.K.O.; formal analysis, Y.N.D.; investigation, V.I.V., N.L.I., E.A.S. and M.L.O.; All authors have read and agreed to the published version of the manuscript.

**Funding:** The work was financial supported by the Russian Foundation for Basic Research (project no. 18-42-520062) and the Ministry of Education and Science of the Russian Federation within the framework of the Russian Federal Targeted Program "Scientific and Scientific–Pedagogical Human resources for the Innovative Russia in 2009–2012" (project No. 2012-1.2.1-12-000-2013-095).

**Acknowledgments:** Authors thank M.A. Bozhenkin, N.A. Alyabina (Lobachevsky State University of Nizhni Novgorod, Russia), L.M. Vinogradskii (Federal Nuclear Center, Sarov, Nizhni Novgorod district, Russia), and T.N. Smyslova (Nizhni Novgorod State Technical Alexeev University, Nizhni Novgorod, Russia) for their assistance in performing the growth experiments; M.N. Drozdov (Institute for Physics of Microstructures, Russian Academy of Sciences, Nizhni Novgorod, Russia) for recording the SIMS spectra, A.N. Tereschenko (Institute for Solid State Physics RAS, Chernogolovka, Moscow distr.) and our colleagues from the Interdisciplinary Resource Center "Nanotechnology," St. Petersburg State University for the help at luminescence measurements.

**Conflicts of Interest:** The authors declare no conflict of interest.

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
