# Peer review of "Low Temperature Growth of the Nanotextured Island and Solid 3C-SiC Layers on Si from Hydric Si, Ge and C Compounds"

_crystals, doi:10.3390/cryst10060491_

Round 1
Reviewer 1 Report
In this manuscript the authors report an analysis of the growth of 3C-SiC islands and layers. In the first part, the morphological evolution is investigated and a nano-crystalline textured structure is recognized. In the second part, luminescence measurements are reported. Finally, in a third part a theoretical modeling of I-V curves and possibly negative differential conductivity in superlattices is reported. The result is however insufficient to deserve publication for several reasons, the most important listed here below:
1. it is not clear which is the achievement and novelty of the present study with respect to the literature as key concepts seem to be already detailed in previous references e.g. [17,18]
2. the language is hardly understandable in many parts because of both poor English (e.g. line 104, "In it "? - line 358-359 "Presence more than one sections with a positive and negative differential conductivity on I-V curve at E > E0 in SL is of principle") and, disappointingly, insufficient care by the authors: wrong labeling and reference to figures (e.g. meaning of the 2-6 spectra in Figure 3b), use of unspecified symbols and acronyms (e.g. FE and BE in page 8 or d1, d2 in eq. 1) or unexplained terminology (e.g. line 117 "the line of the silicon substrate" - line 188 "membrane").
3. the topics of the manuscript are not connected to each other so that the development of the reasoning in the manuscript is not clear
3a. In the first part, the most understandable, different growth stages are discussed but then the compositional analysis by SIMS is apparently looking to a different, unspecified sample (mysteriously named "samples 810" in the caption of Figure 2) including also Ge that was never mentioned before, so that it seems that the sample was obtained by a different growth procedure or that the presence and role of Ge precursors was forgotten before.
3b. Expectedly, the second part about luminescence measurements should consider the samples shown before but surprisingly, the first spectra in figure 3a consider a (111) substrate and this is never mentioned in the text (and should return a different structure/morphology of the growing layers). Also, there are 6 spectra in figure 3b and only the first is identified as the one of Si layer. For the others 2-6 it is just said that they correspond to the "3C-SiC/Si(100) structures" and never explained in the text. Are this different samples at different stages? Different regions?
3c. Part 3 starts telling that the 3C-SiC nanocrystalline layers may form superlattices and then never mention again the material, even in the definition of model parameters. Apparently, this part limits to the general theory that does not offer any insight on the previously reported analysis and is not even specialized to SiC or referred to some experiment. If it has no functional role in the flow of the manuscript, as it seems in the way it is now presented, it has a negative impact and should be removed (and maybe used a starting point for a dedicated study). Moreover, the model presentation is cut too short to be clear. Several symbols in the equations are not explained (d1, d2, \Delta_1, \Delta_2, \delta_0, F_0, E_0, \Omega) and their values are set without any explanation (e.g. saying that \Delta_ij = const. as in line 333 doesn't mean much). Also, there is no comment about the comparison shown in Figure 4. Two parameter sets are considered but there is no reasoning about these choices and the different results.
Comments on more specific points follow:
- The abstract mention X-Ray diffraction but these are never discussed
- According to the present experiments it seems that there is no way to obtain single-crystal structure as the polycrystalline islands nucleate from the very beginning. Is this true or it is expected that single-crystal islands are first formed at earlier growth stages? The authors do not comment on the studies about growth of mono-crystalline 3C-SiC layers (e.g. ref [33], not cited in the text!, or Nishino et al. Appl. Phys. Lett. 42, 460 (1983), Ferro et al. Chemical Vapor Deposition 12, 483 (2006) ).
- The authors state that stress is important and refer to the large 20% lattice mismatch between 3C-SiC and Si. However, there is no clear indication of the residual strain expected in the observed structures, after carbonization and island growth. Is strain going to change at the different growth stages of Figure 1? Is there any indication of strain levels to be inferred from the luminescence analysis? Some useful references that could be cited about the strain relaxation are Long et al. Journal of Applied Physics 86, 2509 (1999), Wen et al. Journal of Applied Physics 106, 073522 (2009)
- The description of the experimental procedure lacks many details, e.g. time or rate of carbonization/growth, indication of the kind of growth reactor, temperature ramps, precursors (no mention to which hydric compounds, silane?), origin of Ge, annealing?
- The figure 1 reports of different growth stages but there is no mention of the time/coverage/thickness of deposition to define such stages. Also the order of panels is confusing. Fig. 1d is a view of an island from panel (a) while figure (c) relates to the coalesced film in panel (b)?
- The cases of Fig. 1e,f are obtained under the same growth conditions of the previous ones (which stage?) and are just imaged in regions where V-lines are present on the surface, acting as preferential nucleation sites? In line 153 a "long annealing" is mentioned? Is it common to all growths? Please specify. Notice the wrong reference to Figure 1d in line 156 and to panel h in the figure caption
- In the paragraph starting at line 256 it is said that the shift in the E2 peak can be related to quantum confinement effect but it is not clear if this is a conclusion of the present study
- In line 270 a comparative analysis between luminescence and SIMS data is referred to but was not discussed before
- In the last paragraph of section 3, which refers to the spectra in Figure 3b (to be explained) it is mentioned that the the BE2_SiC peak corresponds to a layer of composition of 1-2 at.% but this is not clearly explained. Is it coming from the comparison with SIMS?
Reviewer 2 Report
This manuscript reports potentially interesting results on the growth kinetics of the 3C-SiC material over Si substrate. The grown films are analysed at different stage of the processes disclosing some details of the growth mechanism at mesoscopic level. Chemical profiles of elements are also analysed by SIMS. In addition to this morphological analysis the system is investigated in view of photonic or optoelectronic applications. As a consequence PL spectra are also discussed. A theoretical section, with some predictions for the frequency dependent conductivity, is also presented based on the peculiar meso-structures observed. Actually, there is not a close loop where these theoretical results are compared with properties of the real material. Anyhow, apart this missing connection, I believe that the results here presented are sound and of interest for the application of cubic SiC polytype. Moreover, both the processing and characterization results are representative of an intense and complete activity. As a consequence I am in favour with the publication of the paper in MDPICrystals after a mandatory link between the model parameters and the properties of the fabricated system. In addition, I also notice some weaknesses in the manuscript form which also need improvements.
Firstly, the manuscript needs a careful and sever proofreading since several misprints appears. E.g. it is surprising to find a compilation note (see section 4. Title) in a submitted manuscript!
The descript of the growth process is incomplete. The applied process recipe should be carefully presented in order to permit the reproduction of the data. For example, the duration of the carbonization step is not indicate as the partial pressure of the components in the CVD mixtures.
Connected to the previous comment, a puzzling experimental evidence is the presence of Ge. This inclusion of this element does not appear in the fabrication steps, and its presence is not “natural” as the O one. There is a sentence “Despite that germanium was fed to the growth chamber during the whole growth” which can explain the Ge presence but I do not see any Ge citation in the part dedicated to the fabrication of the film. Also the presence of the oxygen should be justified since its level seems above the usual accepted contamination level.
The introduction justifies the 3C-SiC interest only for the light emitting properties. I would suggest to extend this vision. Actually 3C-SiC is an interesting electronic material for sever power electronics applications if high quality (defect free) films will be produced (see e.g. Appl. Phys. Rev. 7, 021402 (2020) and reference therein). I believe that the present research could have a good impact (e.g. the observed texturing) also for the community working on power electronic application of cubic SiC.
Reviewer 3 Report
The article “Low Temperature Growth of the Nanotextured Island 3 and Solid 3C-SiC Layers on Si From Hydric Si, Ge 4 and C Compounds Text in introduction should be improved for clarity” represents interesting results on Different growth stages and surface morphology of the epitaxial 3C-SiC/Si(100) structures. The article could be accepted if major revision of text and grammar and interpretations is performed. Changes that should be addressed:
Lines 24-25: sentence unclear,
Line 42: unclear sentence “elevated irradiative recombination activity”, clarifications and references needed,”
Line 104: improve, clarify “In it, we study the surface morphology”
Lines 112-113: ref neded eg in the sentence “The growth temperature determines the disintegration rate of the hexane and silane molecule adsorbed by growth surface and consequently a number of silicon and carbone adatoms participating in a growth process.”
“Carbone” - “carbon”
Line 156 The groove occupation by the adsorbed atoms provides a complex profile of the linear growth figure along its guide.
Line 165: “depth greater then” - than
Line 168: “in the vicinity of”
Line 196: References about 3C light emitting devices should be added Phys. Scripta T148, 014002, 2012.
Line 224: “was studied many investigators” - “was studied by many investigators”, additional eferences should be added here: J. Electron Mater. 40, 394, 2011, J. Lumin 134, 588, 2013.
Line 235 “typical for cubic silicon carbide” - reference for the luminescence, J. Lumin 134, 588, 2013
It must be mentioned that higher PL energy bands can be due to inclusions of 6H and 4H SiC as growth islands (Mater. Sci. Forum 711, 159, 2012) and also as stacking faults (J. Cryst. Growth, 395, 109, 2014).
Line 297: after “Discussion” comes unclear sentence in brackets which should be improved.
“being possible heir a quantum superlattice”
Line 310: sentence unclear “If earlier it was the multilayer structures with a one-dimensional SL that were mostly investigated, in recent years two-dimensional and three-dimensional SL’s have also come into considering.”
Liner 313: unclear: “That is why may wait that the presence of an additional superstructure in a crystal layer”
Line 329: “to reveal some specific Presence more than one sections”
Line 353: correct that “ [41] have been find”
Fig4 caption has to be corrected, “мэВ” should be correted, and too many parameters indicated
Line 358: sentence “Presence more than one sections with a positive and negative differential conductivity on I-V curve at E > E0 in SL is of principle [35]. In this situation, there is a chance to realize in a system the negative dynamic conductivity at absent of low frequency self-excited oscillation [40].“ has to be corrected.
Reviewer 4 Report
In this manuscript, the authors discuss the nucleation and expansion of 3C–SiC islands on the carbidized silicon surface and a formation of the continuous 3C-SiC layer during a low temperature epitaxial process. I really enjoy reading the whole manuscript and like the content presented; therefore, I would like to suggest “minor revision” for its publication in crystals. Here come some of my comments:
1. Except for lighting applications, 3C-SiC has been used for high-Q resonators and might be potential for 5G applications. Please include the following papers in the introduction section.
2. In Figs. 1, there are some confusion for their figure captions.
3. Can the authors measure the rocking curves of 3C-SiC layer on Si? It would be more straightforward for readers to understand how well the 3C-SiC layer is grown on Si.
Round 2
Reviewer 1 Report
In the revised version of the manuscript, additional technical details have been included but no change has been done in the explanation, discussion and arrangement of the content. No attempt has been done to address the major problems highlighted in the first negative review so that the evaluation is still the same: rejection. I do not want to repeat comments but I underline the key points that the authors did not manage to correct:
- when talking about the connection between the three parts of the paper, i.e. growths, optical measurements and superlattices, there is no doubt that they are connected in general and of course I had no criticism about that. The problem is that here, inside the manuscript, the three parts are not integrated in a unique storyline. First, some samples are shown, then other samples with totally different parameters are analyzed and finally a purely theoretical section without any reference even to the material is reported. As a reader, it seems that you are just combining things without considering any flow of reasoning and hence it is hard to understand why you pass from one to the other. Probably, the first two parts could compose, after a major rewriting of the manuscript (that was expected from previous comments) an quite understandable scenario. The perspective of quantum effects from superlattice structure is surely fascinating and I encourage you to pursue that target but it is nonsense to spend a full section just talking of perspectives without any direct reference to the system or any proof. Mentioning that "this structures could be used as superlattices, paving the way for new quantum effects.... to be treated elsewhere" would attract the same interest. In addition, the section is not at all clear and (as you say) a more detailed description would fall out of the scope of the manuscript and journal. The fact that the corresponding Figure 4 is reported and not really commented in the text (not even distinguishing and discussing the different content of panel a and panel b) testifies that this part lacks of explanation and would be inexplicable to the average reader, unless very expert in that topic.
- as indicated in the previous review, achievements and novelty were not clear from the text and, since no changes has been made, the problem is still there. You wrote a list of points in the answer but these are not clearly stated in the manuscript. Moreover, some of them seem to be already discussed in previous publications of yours. E.g. in Figure 5 of ref [18] it seems that you already show textured 3C-SiC so if you claim this is novel in the present manuscript you should mention what is the difference from ref [18].
- After your explanation of Figure 1, it seems that you are not really considering different growth stages but different growth temperature. I think you cannot say this is equivalent as changing temperature modifies several phenomena (e.g. diffusion) that could have a great impact on the surface morphology. Growth stages should be investigated by repeating the same experiment for different duration. If you can demonstrate that changing temperature is exactly the same of considering different growth times, please explain
- the English level did not improve. Also some replies are hardly understandable and even some of the few changes are still not correct, e.g. "It is principle if the current-voltage characteristic of the SL contains more than one curve lines with a positive
and negative differential conductivity". The text should be entirely reviewed with this respect
In addition, there are still many obscure details that are hardly understandable or barely motivated, e.g. what is the point of having 6 lines in the Figure 3b if not commenting about them (line 2 vs. 4?)
Author Response
Thank you for your comments. Please check the attachment.

Reviewer 3 Report
Modeling description is not clear and not supported by experimental verification.
Author Response

(The authors gave the same response as above.)
